# A qPCR technology for direct quantification of methylation in untreated DNA

Kamilla Kolding Bendixen [1,2] ✉, Maria Mindegaard [1], Samantha Epistolio [3], Giulia Dazio[3], Francesco Marchi [4], Paolo Spina [3,5], Eva C. Arnspang[6], Mette Soerensen [2], Ulf Bech Christensen [1], Milo Frattini[3,7] & Rasmus Koefoed Petersen[1,7]

DNA methylation is important for gene expression and alterations in DNA methylation are involved in the development and progression of cancer and other major diseases. Analysis of DNA methylation patterns has until now been dependent on either a chemical or an enzymatic pre-treatment, which are both time consuming procedures and potentially biased due to incomplete treatment. We present a qPCR technology, EpiDirect®, that allows for direct PCR quantification of DNA methylations using untreated DNA. EpiDirect® is based on the ability of Intercalating Nucleic Acids (INA®) to differentiate between methylated and unmethylated cytosines in a special primer design. With this technology, we develop an assay to analyze the methylation status of a region of the *MGMT* promoter used in treatment selection and prognosis of glioblastoma patients. We compare the assay to two bisulfite-relying, methyl-specific PCR assays in a study involving 42 brain tumor FFPE samples, revealing high sensitivity, specificity, and the clinical utility of the method.

DNA methylation is an epigenetic modification of the DNA, where especially the methylation of cytosine, occurring by the addition of a methyl group to the fifth carbon of cytosine creating 5-methylcytosine (5mC), is of interest. Abnormal alterations in DNA methylation patterns have been associated with the development of cancer and other human diseases[1–3]. Methylation patterns of specific genes have been shown to have clinical significance with relevant prognostic, diagnostic, and predictive roles. Current clinically relevant methylation markers within cancer include *MGMT*, *MLH1*, *GSTP1*, *SEPT9*, and *LINE-1*[4,5]. O[6]-methyl-guanine-DNA methyltransferase-encoding gene (*MGMT*) is used as a predictive marker for response to temozolomide (TMZ) in patients with gliomas, where hypermethylation of the *MGMT* promoter is associated with increased efficacy of TMZ[6,7]. *MGMT* methylation is also a prognostic marker; patients with hypermethylated *MGMT* have been found to have a better prognosis irrespective of the choice of treatment[6].

The analysis of DNA methylation remains a challenging task. DNA methylation cannot be analyzed directly using standard PCR approaches because the methylation patterns are not maintained during the amplification. The most utilized methods for methylation analyses overcome this issue by making a permanent chemical conversion of purified DNA using a sodium bisulfite treatment prior to PCR. The sodium bisulfite treatment converts unmethylated cytosines to uracils while leaving 5mC unchanged. This creates a sequence difference between unmethylated and methylated DNA, that is easily analyzed using approaches like methyl-specific PCR (MSP)[8] or pyrosequencing[9]. Other approaches include qMSP and methyl-sensitive high-resolution melt allowing for relative quantification of the methylation status in a PCR cycler[10,11]. High differences in recovery, conversion efficiency, and conversion specificity of bisulfite have been observed in different kits from various manufacturers[12–15], potentially leading to differences in

[1]PentaBase A/S, Odense, Denmark. [2]Epidemiology, Biostatistics and Biodemography, Department of Public Health, University of Southern Denmark, Odense, Denmark. [3]Laboratory of Molecular Pathology, Institute of Pathology, Ente Ospedaliero Cantonale (EOC), Locarno, Switzerland. [4]Service of Neurosurgery, Neurocenter of the Southern Switzerland, Regional Hospital of Lugano, Lugano, Switzerland. [5]Department of Health Sciences, University of Eastern Piedmont, Novara, Italy. [6]Department of Green Technology, University of Southern Denmark, Odense, Denmark. [7]These authors contributed equally: Milo Frattini, Rasmus Koefoed Petersen. ✉e-mail: kkb@pentabase.com

precision between kits. Lastly, bisulfite treatment is harsh on the DNA, causing degradation of the DNA, which especially can pose a challenge when having small biopsies, e.g., formalin-fixed paraffin-embedded (FFPE) biopsies from brain tumor or cell-free DNA from liquid biopsies. Hence attempts have been made to develop bisulfite-free analysis methods.

One of the most applied bisulfite-free methods involves enzymatic digestion by methylation-sensitive restriction enzymes (MSREs)[16,17] followed by a methyl-specific-multiplex ligation-dependent probe amplification (MS-MLPA)[18]. The use of MSREs is less harsh on the DNA compared to bisulfite treatment. However, the MSREs are sequence-dependent, limiting this test for methylation evaluation only to some regions containing the restriction sites. Newer methods include TET-assisted pyridine borane sequencing (TAPS) which allows base-level sequencing of the DNA[19], by enzymatic oxidation and reduction by pyridine borane creating dihydrouracil, which will be read as thymine during sequencing. The other method is Enzymatic Methyl-seq (EM-seq)[20] which applies only enzymatic steps to deaminate cytosine while 5mC remains unchanged. However, all these methods are time-consuming and require multiple hands-on steps.

In this paper, we use the physiochemical properties of 5mC to develop a qPCR platform for the analysis of DNA methylation without any prior treatment of the DNA. We do so by employing the DNA analog platform called Intercalating Nucleic Acid (INA®) affecting the π-stacking of DNA[21]. In short, double-stranded DNA is stabilized by Watson-Crick base-pairing and by base-stacking, where the latter has been shown to be the main stabilizer of the DNA double helix[22]. The base-stacking is the effect of the interactions between the electrons of π-electrons, and thereby also named π-stacking. 5mC has been shown to influence the thermal stability of DNA duplexes even though it does not participate in the Watson-Crick base-paring[23,24]. INA® comprises a minimum of one nucleobase analog, called an intercalating pseudo-nucleotide (IPN), composed of a flat, conjugated aromatic or hetero-aromatic ring system, linked to the phosphodiester backbone at fixed positions of a synthetically manufactured oligonucleotide. The IPN will coaxially stack with the neighboring nucleobases and thereby increase the π-stacking energy of the DNA helix, which increases the melting temperature $(T_m)$[21]. The properties of the INA® have among other utilizations been used for sensitive detection of single-point mutations in DNA[25].

In this work, we present a property of the INA®, an apparent difference in the stacking effect due to the moieties of INA® stacking differently with 5mC compared to cytosine, reflected as an enhanced thermal stability of methylated DNA over non-methylated DNA. We use this effect in a qPCR platform to directly detect and quantify DNA methylation without any pre-treatment of the template DNA. We call the method used in this paper EpiDirect®. EpiDirect® utilizes a special primer design, called an EpiPrimer™ that allows selective amplification of methylated DNA over unmethylated DNA.

## Results

### A platform for direct methylation quantification
We have developed a PCR platform, EpiDirect®, allowing for direct quantification of DNA methylation without any pre-treatment of the DNA. With this platform, we utilize the synergistic effect between cytosine methylation and the DNA technology, INA®.

An EpiPrimer™ consists of three parts: An anchor-, a loop-, and a starter sequence, as illustrated in Fig. 1. The anchor sequence comprises IPN molecules and covers the targeted CpG area. The anchor sequence anneals to a complementary methylated target with significantly higher thermal stability compared to a complementary unmethylated target. The starter sequence is placed in the 3′-end of the primer and is designed to have low thermal stability and should therefore not bind to the template DNA if it does not get support from

the anchor sequence. As the anchor sequence will bind with higher thermal stability to methylated DNA, it is possible to make selective amplification of methylated DNA by performing the PCR at an annealing temperature higher than the $T_m$ of the anchor sequence bound to unmethylated DNA.

To be able to make relative quantification of the methylation status of a sequence, an independent reference sequence is amplified simultaneously but is detected in a different fluorescence channel. The reference assay is designed in a gene unaffected by methylation status. If threshold is reached at approximately the same cycle during PCR, as shown in Fig. 1a, (the ΔCt value between the methyl-specific assay and internal reference assay is low), the target DNA sequence is highly methylated in the sequence of interest. If there are no CpG methylations in the targeted sequence, the EpiPrimer™ will have a very low affinity towards the DNA and thereby causing a high ΔCt value, as illustrated in Fig. 1b.

The methylation pattern will only be present in the DNA from the original sample and not in the amplicons made during PCR by the polymerase. This also means that the anchor part of the EpiPrimer™ has a low affinity towards the amplicons. Therefore, we have introduced a loop sequence, connecting the anchor and starter sequence, to regain affinity for sequences originally comprising DNA methylations. Furthermore, the polymerase cannot read over the IPN molecules, and therefore, the anchor part of the EpiPrimer™ will not be replicated.

### Intercalating nucleic acids have increased thermal stability to DNA
We investigated the influence of fluorescent INA® probes compared to DNA probes (herein called Ref) on the thermal stability of complementary oligonucleotides. We evaluated two different IPN molecules (IPN 1 and IPN 2, see Fig. 2a for structures) coupled into a 14-nucleotide sequence of the *MGMT* promoter 5′ to the G of the CpG sites 5′-**CG**TCC**CG**AC**G**CC**CG**-3′. We denote the probe modified with IPN 1 for INA-1 and the probe modified with IPN 2 for INA-2. The IPN molecules 1 and 2 were respectively coupled into the sequence at each of the four CpG sites and hybridized to a complementary unmethylated (UM) target creating a probe/target duplex. The $T_m$ increased by 7.3 °C utilizing INA-1 and 7.8 °C for INA-2 compared to Ref. The $T_m$ values are shown in Fig. 2b.

### Methylated CpG sites increase the thermal stability of DNA
The Ref probe was hybridized to UM and a fully methylated (M) target and heated to obtain the $T_m$, see Fig. 3a. The $T_m$ was 64.6 ± 0.03 °C and 68.9 ± 0.08 °C when hybridized to target UM and M, respectively. Hence, the four methylated CpG sites increased the $T_m$ by ~4.3 °C in total compared to unmethylated CpG sites (Fig. 3b).

### Synergistic effect of intercalating nucleic acids and DNA methylation
We investigated the influence of DNA methylations in the target sequence on the melting properties of the INA® probes (Fig. 4a). As shown in Fig. 4b, INA-2 bound to target M melted 11.0 °C higher than INA-2 bound to target UM. Hence, the use of INA-2 compared to Ref increased the $\Delta T_m$ by 6.6 °C. Utilizing INA-1 increased the $\Delta T_m$ by 4.2 °C compared to Ref, see Fig. 4c.

### Correlation between the number of methylated CpG sites and duplex stability
The addition of the first methyl group to any of the cytosines in the *MGMT* promoter sequence increased the affinity towards the complementary Ref probe by 2.1–2.9 °C depending on the position of the methyl group. Using INA-2, the comparative affinity was increased by 5.3–5.7 °C. The second to fourth addition of methylation groups further increased the affinity by ~0.6 °C and ~1.9 °C per 5mC for Ref and

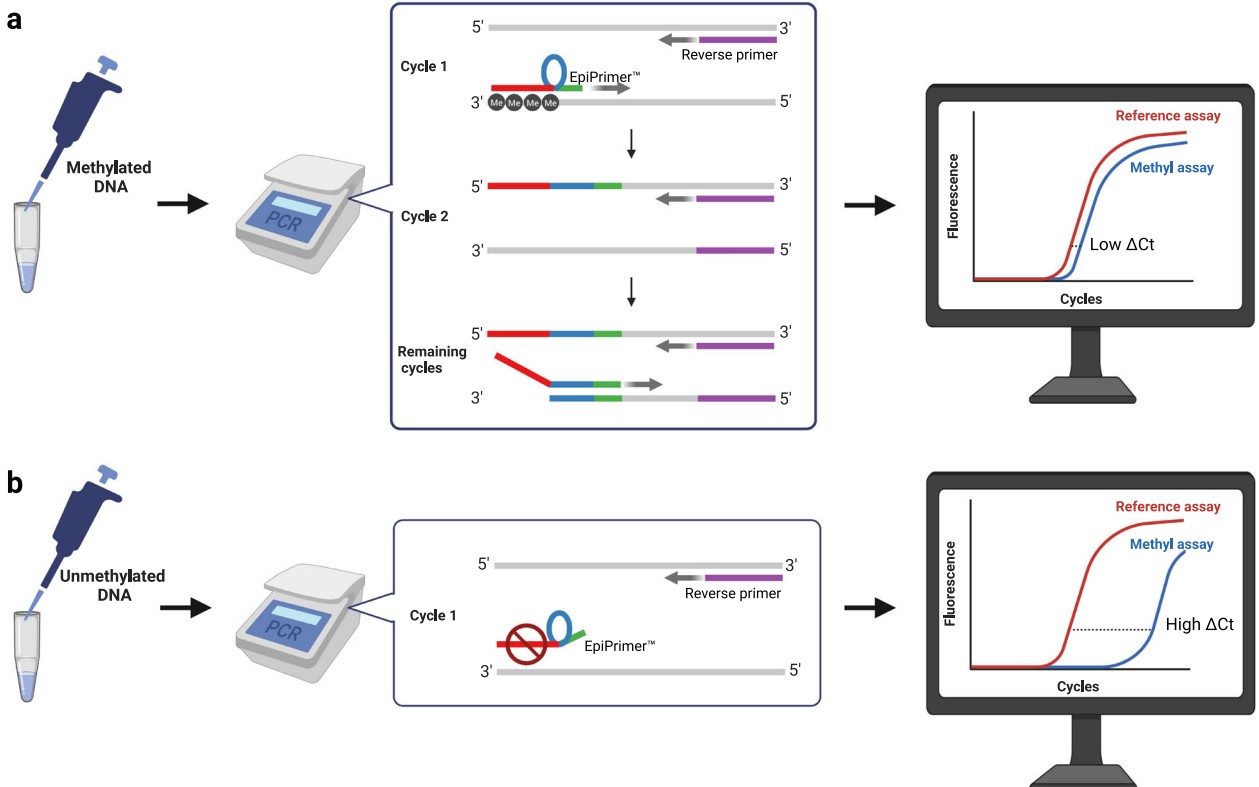

**Fig. 1 | Principle of the EpiDirect® platform for selective amplification of methylated DNA.** The methyl-specific assay (methyl assay) utilizes an EpiPrimer™ that consists of three parts: An anchor sequence (red), a loop sequence (blue), and a starter sequence (green). The anchor sequence is based on the Intercalating Nucleic Acid (INA®) technology and covers the sequence comprising the CpG sites of interest. **a** The proposed mechanism of amplification of untreated methylated DNA is illustrated. In the first cycle, the anchor sequence will bind with high thermal stability to methylated DNA allowing for the annealing of the starter sequence and initiation of the PCR. In cycle two, the reverse primer will bind to the amplicon created with the EpiPrimer™, but the polymerase will not replicate the anchor sequence as it contains Intercalating Pseudo-Nucleotide (IPN) molecules. In the

remaining cycles, the loop and starter sequence will work as a regular primer that amplifies the amplicons together with the reverse primer. Compared to the internal reference assay (reference assay) with an efficiency close to 100%, the EpiPrimer™ is similar or a little less efficient on a methylated template. This gives zero or a small difference in the Ct values between the methyl-specific assay and the internal reference assay. **b** The workflow of analysis if the target DNA is unmethylated DNA is illustrated. If there is no DNA methylation in the CpG sites of the anchor sequence's target region, the EpiPrimer™ will only bind weakly to the DNA and thereby causing a large difference in the Ct values between the methyl-specific assay and the internal reference assay. Created with BioRender.com.

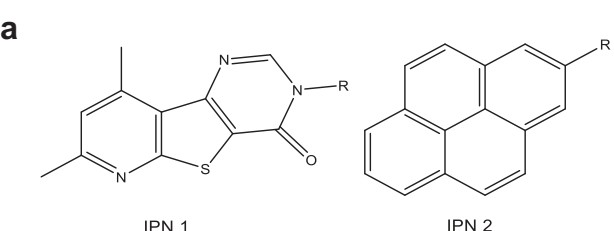

**a**

IPN 1          IPN 2

**b**

| Probe | Mean $T_m$ (°C) | +$T_m$ (°C)[a] | 95% CI |
|-------|-----------------|----------------|--------|
| Ref   | 64.6 ± 0.03     | -              | -      |
| INA-1 | 72.4 ± 0.01     | 7.8            | [7.8; 7.9] |
| INA-2 | 71.8 ± 0.25     | 7.3            | [6.7; 7.9] |

[a] +$T_m$ = $T_m$(INA) - $T_m$(Ref)

**Fig. 2 | Intercalating nucleic acid (INA®) increases the melting temperature of DNA oligonucleotide duplexes. a** The skeletal structure of intercalating pseudo-nucleotide (IPN) type 1 and type 2. R denotes a linker sequence, which binds the molecule to a phosphoramidite allowing for coupling into oligonucleotides. IPN molecules are composed of several conjugated double bonds and are thereby adding π-electrons to the DNA double helix. This increases the overall π-stacking

energy of the DNA. **b** Table of the mean melting temperature ($T_m$) and standard deviation (SD) of fluorescent oligonucleotide probes bound to an unmethylated complementary target (mean ± SD. $n$ = 3 technical replicates). Placing IPN 1 and IPN 2 5' to the G of the CpG in the probe sequence (creating INA-1 and INA-2, respectively) increased the $T_m$ of the DNA duplex compared to the regular DNA probe (Ref). Source data are provided as a Source Data file for Fig. 2b.

INA-2, respectively. Linear regressions are seen in Fig. 4d. All $T_m$ values can be found in Supplementary Table 1.

## EpiDirect® MGMT Methylation qPCR Assay verification

The EpiDirect® MGMT Methylation qPCR Assay had a linear dynamic range from 100% to 2.5% methylation. A linear fit to the data points gave the equation $y = 6 - 2.9x$, where $y$ is the ΔCt and $x$ is log$_{10}$ to the percentage of methylation. The fit gave a R² value of 0.976, see Fig. 5a.

The percentage methylation can thus be calculated from the ΔCt value by Eq. 1.

$$\%methylation = 10^{\frac{\Delta Ct - 6}{-2.9}} \qquad (1)$$

The assay was found to have similar linear dynamic range with input DNA concentrations of 0.1–10 ng/µL sample, corresponding to

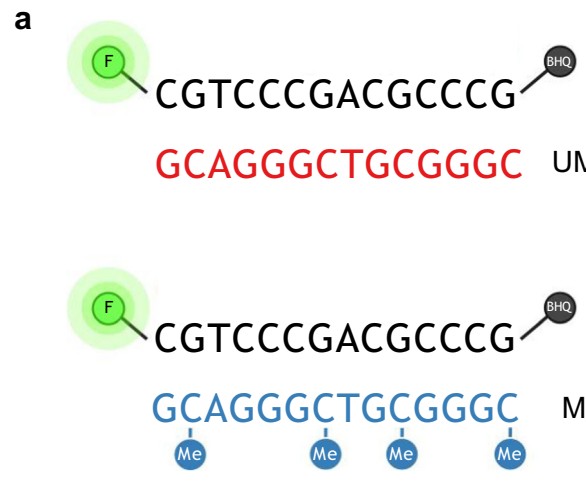

**Fig. 3 | Thermal stability is influenced by methylation. a** Sequences of the oligonucleotides used for the study. A fluorescent probe was synthesized with a FAM™ fluorophore (F) in the 5′-end and a black hole quencher 1 (BHQ®-1) in the 3′-end. The probe covered a sequence found in the promoter region of *MGMT* comprising four CpG sites. The probe was hybridized to a complementary oligonucleotide being either unmethylated (UM) or methylated at all four cytosine nucleobases (M). Created with BioRender.com. **b** Thermal stability of the two duplexes. The unmethylated DNA duplex (red) had a melting temperature ($T_m$) of $64.6 \pm 0.03$ °C and the methylated complex (blue) melted at $68.9 \pm 0.08$ °C giving a difference of 4.3 °C (95% CI [4.2; 4.5]).

an input of 0.5–50 ng. The PCR efficiency in this range was 97.2% in both the FAM™ and HEX™ channels.

The limit of blank (LOB) was calculated to be ΔCt 5.0, corresponding to 2.3% methylation. From this, the technical limit of detection at 95% certainty (LOD) was calculated to be ΔCt 4.9 giving 2.4% methylation. The LOD was tested at 3% methylation. Nineteen out of twenty replicates were below ΔCt 4.9 meaning the experimental LOD was 3% methylation. The assay's methylation calling from the twenty replicates was average $3.4 \pm 0.78$%. All twenty replicates of unmethylated DNA purified from FFPE material were above the LOD of ΔCt 4.9 and were therefore correctly called unmethylated. Nineteen out of twenty replicates of 3% methylated DNA diluted into DNA purified from FFPE material were detected as being methylated. The average percentage calling of these replicates was $3.9 \pm 1.4$%. The data is shown in Fig. 5b.

### EpiDirect® MGMT Methylation qPCR Assay validation

We compared the methylation calling of EpiDirect® MGMT Methylation qPCR Assay with two comparator methods: A methyl-specific PCR assay followed by gel electrophoresis (comparator method 1) and quantitative real-time MSP Assay (comparator method 2) using 42 FFPE samples from brain tumors.

In total 31 of 42 samples had the same results in all three methods; six samples were analyzed to be methylated and 25 samples were unmethylated by all three methods, illustrated in the center of the Venn diagrams in Fig. 6a, b, respectively. The remaining 11 samples displayed agreement in two methods, specifically, five samples displayed agreement between EpiDirect® and one of the other methods. EpiDirect® MGMT Methylation qPCR Assay found additionally five samples to be methylated, which were determined to be unmethylated by the two comparator methods. These five samples ranged from 2.4% to 6.5% methylation according to the EpiDirect® analysis. A visual overview of the percentage methylation calling by EpiDirect® and comparator method 2 is illustrated in Fig. 6c.

The sensitivity of EpiDirect® MGMT Methylation qPCR Assay, setting comparative method 1 as a reference, was 0.82 (CI 95% [0.52; 0.95]) and the specificity was 0.84 (CI 95% [0.41; 0.93]). Setting comparative method 2 as the reference, the sensitivity was 0.75 (CI 95% [0.36; 0.96]) and the specificity was 0.76 (CI 95% [0.60; 0.88]). The two comparative methods found the same seven samples to be methylated and 30 samples to be unmethylated and, thereby, had agreement in 88.1% of the samples. Setting these samples as a combination reference, the sensitivity of EpiDirect® MGMT Methylation qPCR Assay was 0.86 (CI 95% [0.48; 0.97]) and the specificity was 0.83 (CI 95% [0.66; 0.93]).

### Discussion

We have presented a platform for the quantification of methylation in untreated DNA solely using qPCR technology. In this study we found an increase in $T_m$ going from zero methylation sites to a single methylation by 2.1–2.9 °C in our non-INA®-modified set-up, which was higher than Nardo et al. who found an increase of ~1.6 °C[23]. Increasing the number of methylation sites from one to four increased the $T_m$ by ~0.6 °C per additional methylation, which was also higher than what was found by Nardo et al. (~0.3 °C per methylation). A study by Tsuruta et al. reported an increase in $T_m$ of ~1 °C per methylation[24]. The buffer that is utilized for melt studies can affect the $T_m$ and the $T_m$ is highly dependent on the sequence, which can explain the variations observed between these studies. However, we can confirm a correlation between the $T_m$ and the number of cytosine methylation in the sequence. Another observation is that both our study and the one by Nardo et al. found that the $T_m$ was more affected by going from no methylations to one methylation compared to the addition of further methylation sites, even though we used different set-up and sequences[24]. Using our INA® technology it was possible for us to increase the $T_m$ by up to 5.3–5.7 °C with a single methylated cytosine in our target sequence compared to an unmethylated target and 1.9 °C per additional methylation sites in the target sequence, which, to the best of our knowledge, has not been reported previously.

Beside the advantages aforementioned, we have noticed some limitations concerning the PCR conditions of the EpiDirect® assay. The *MGMT* sequence used in this study requires the annealing temperature of 76 °C to have the highest discrimination between the methylated and unmethylated targets. In addition, the anchor sequence that anneals to the CpG sites of interest is 14 nucleotides long; if it was longer or had a higher CpG content, the annealing temperature of the PCR would be even higher. We could partly overcome the issue of extreme PCR conditions by including a few mismatches in the anchor sequence, which lowers the annealing temperature. In EpiDirect®

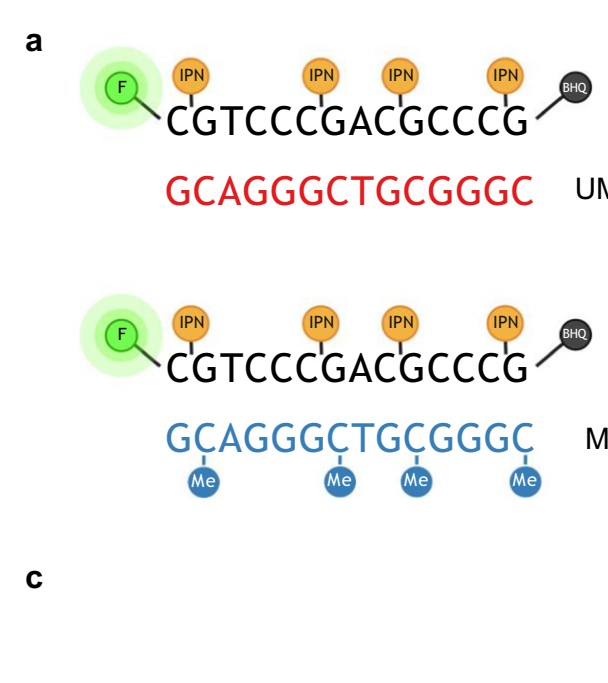

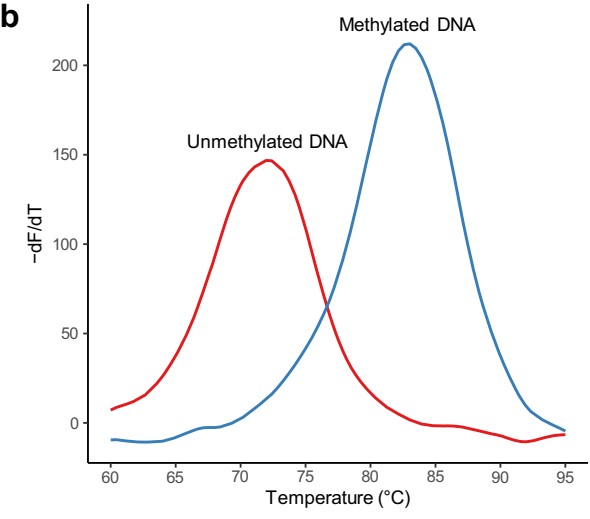

| Probe | Mean $\Delta T_m$ (°C)[a] | +$\Delta T_m$ (°C)[b] | 95% CI |
|--------|--------|--------|--------|
| Ref | 4.3 ± 0.1 | - | - |
| INA-1 | 8.6 ± 0.1 | 4.2 | [4.0; 4.5] |
| INA-2 | 11.0 ± 0.5 | 6.6 | [5.5; 7.7] |

[a] $\Delta T_m = T_m(M) - T_m(UM)$
[b] +$\Delta T_m = \Delta T_m(INA) - \Delta T_m(Ref)$

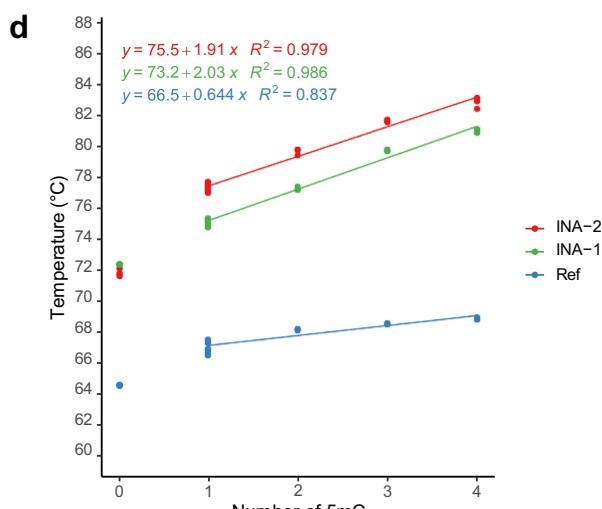

**Fig. 4 | Combined thermal effect of intercalating nucleic acid (INA®) and CpG methylations. a** Sequences of the oligonucleotides used for the study. A fluorescent probe was synthesized with a FAM™ fluorophore (F) in the 5′-end and a black hole quencher 1 (BHQ®-1) in the 3′-end. The probe covered a sequence of four CpG sites in the *MGMT* promoter. Intercalating pseudo-nucleotide (IPN) molecules, either type 1 or type 2, were coupled 5′ to the G of the CpG sites, creating probe INA-1 and INA-2, respectively. The probes were hybridized to a complementary unmethylated oligonucleotide (UM), and a target methylated at all four CpG sites (M). Created with BioRender.com. **b** Melt peaks of the methylated (blue, 82.8 ± 0.36 °C) and unmethylated (red, 71.9 ± 0.25 °C) targets hybridized to probe INA-2, giving a

difference of 11.0 °C (95% CI [10.2; 11.7]). **c** Table of the mean melting temperature differences ($\Delta T_m$) and standard deviation (SD) of fluorescent oligonucleotide probes bound to unmethylated (UM) or fully methylated (M) complementary target, respectively (mean ± SD. $n = 3$ technical replicates). INA-1 and INA-2 increased the $\Delta T_m$ compared to Ref. **d** Linear regression on the number of methylated CpG sites (one to four sites) and melting temperature of DNA duplexes using probe INA-2 (red), INA-1 (green), and Ref (blue). A linear increase in the $T_m$ by -1.9 °C for INA-2, -2.0 °C for INA-1, and -0.6 °C for Ref per extra methylation in the target (one to four methylations) was found. Source data are provided as a Source Data file for Fig. 4c and 4d.

MGMT Methylation qPCR Assay we have included a single mismatch to reduce the annealing temperature. Without this mismatch, the optimal annealing temperature was around 80 °C, which possessed challenges to the PCR design.

The clinical validation of the assay revealed some variance between the methods, especially in the cases where the methylation degree was low (<10%) according to EpiDirect® MGMT Methylation qPCR Assay and/or comparator method 2, which also gave semi-quantitative results. The two comparator methods found the same seven samples to be methylated, and EpiDirect® MGMT Methylation qPCR Assay found six of these samples to be methylated, and one sample was therefore false negative. This sample was called 1.8% methylated by comparator method 2, and it is likely that it was below the limit of detection of EpiDirect®. The discordances could be due to several reasons, first of all, comparator method 1 and EpiDirect® do not

analyze the same CpG sites, and, in addition, the CpG sites analyzed in method 2 are unknown. Secondly, EpiDirect® used a cut-off corresponding to the technical LOD of 2.4% methylation, and comparator method 2 used >0.6% as the cut-off whereas comparator method 1 detected methylation by visual inspection of the gel and no cut-off was therefore set in terms of percentage of methylation. However, we must keep in mind that there is no global consensus on which cut-off is optimal to predict the efficacy of TMZ and the prognosis. A survey published in 2019 found that the cut-off from several laboratories varied from 3% to 30% methylation using pyrosequencing[26]. The bisulfite treatment has been shown to be affected by several factors related to the DNA itself like quantity and quality[27,28], which could explain some of the variances in the optimal cut-off. Finally, due to the high severity of the disease, most patients are substantially treated with the same protocol (usually, the Stupp protocol), regardless of the

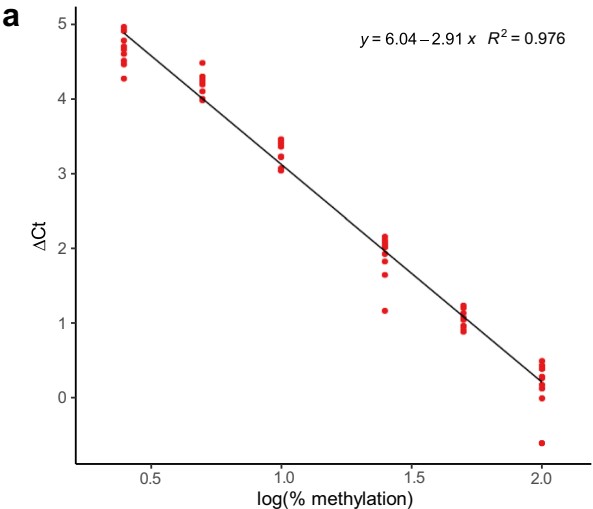

**Fig. 5 | Performance studies from the EpiDirect® MGMT Methylation qPCR Assay. a** Linearity was observed from 100% to 2.5% methylation. The ΔCt values are plotted as a function of the logarithm to the percentage of methylation of the DNA template. Each sample was evaluated in 10 replicates from multiple PCR runs. **b** Experimental data for 20 replicates of unmethylated and 3% methylated DNA. The unmethylated DNA was evaluated using DNA purified from whole blood (red) and FFPE material (blue). 3% methylated DNA (green) and 3% methylated DNA diluted in DNA from FFPE material (purple) were evaluated. Nineteen out of twenty replicates were detected for the 3% methylated DNA samples. The technical limit of detection (LOD) is plotted as a broken line at ΔCt = 4.9. Five ng of DNA was used as input for each replicate. Source data are provided as a Source Data file for Fig. 5a and 5b.

status of *MGMT* promoter methylation[29], this parameter is mainly used for predicting the duration of the efficacy of the treatment.

To conclude we observed that the introduction of EpiDirect® platform in diagnostic routine could be relevant as it saves valuable time in the laboratory and overcomes the risk of incomplete or over-conversion of the DNA during the bisulfite treatment, which could cause inaccurate estimation of the methylation percentage. The EpiDirect® platform can also be used to detect methylation on other target genes, but this is not the focus of the present paper. This platform opens a potential for wider use of methylation analysis allowing smaller laboratories with limited resources to perform the methylation analysis themselves as well.

## Methods

### Ethical statement
The validation study had ethical approval with the approval code Ref. EC 3721 - BASEC 2020-01939 (16th October 2020) with ethics over-sights by Comitato etico cantonale, Via Orico 5, 6501 Bellinzona CH. Informed consent was obtained, and patient compensation was not provided.

### *MGMT* region for the study
A sequence comprising four CpG sites located in exon 1 of the *MGMT* promoter was used for the melt studies and qPCR assay. The CpG sites were located on chromosome 10, GRCh38.p13 from 129,467,250 to 129,467,263 "**CG**TCC**CG**A**CG**CC**CG**". These CpG sites are numbered 75 to 78 (numbering according to the study by Malley et al.[30]) and are associated with survival for patients with glioblastoma[31,32].

### Design of fluorescent oligonucleotides for melt studies
Three different oligonucleotides were synthesized covering CpG sites no. 75 to 78 of the *MGMT* promoter. The oligonucleotides were synthesized as fluorescent probes with a FAM™ fluorophore in the 5′-end and a black hole quencher 1 (BHQ®−1) in the 3′-end. One probe was based on standard DNA chemistry (Ref, sequence ID: P1) and two other probes were modified by IPN molecules. Two probes were synthesized with the IPN molecules coupled 5′ to the G of the CpG sites using either IPN 1, creating probe INA−1, ID: P2, or IPN 2, creating probe INA-2, ID: P3. All oligonucleotide sequences can be found in Table 1.

### Design of target oligonucleotides for melt studies
Eight targets complementary to sequence ID P1-P3 were designed: One oligonucleotide having unmethylated cytosines (UM, ID: T1), and one oligo having 5mC amidites at all four CpG sites (M, ID: T2). 5-Methyl-deoxy Cytidine (n-bz) CED phosphoramidite (ChemGenes Corporation, Wilmington, MA, US) was used as amidite for 5mC. We investigated the influence of the number of methylated CpG sites and the methylation pattern on the T_m of the DNA duplex utilizing the probes ID: P1-P3 and target ID: T3-T8. Four targets were designed with one methylated CpG at either of the four CpG sites in the *MGMT* target sequence (ID: T3-T6). Furthermore, we synthesized a target with two and three methylated CpG sites (ID: T7 and T8) to investigate the correlation between the number of methylated CpG sites and the T_m.

### Protocol for melt studies
Probe sequence ID: P1-P3 were hybridized to the complementary targets with various CpG methylation patterns, ID: T1-T8, and heated to obtain the T_m of the duplex. One probe sequence was mixed with one target sequence in each tube, and the T_m was measured in triplicates. Twenty µL buffer containing 0.02 M Na_2HPO_4, 0.02 M NaCl, and 2 mM EDTA (in short TM buffer) was added to PCR tubes with 2.5 µL (giving a working concentration of 2 µM) of the probes and 2.5 µL of the targets (4 µM). The mixtures were melted in a BaseTyper48.4 Quiet HRM Real-Time PCR System (PentaBase A/S, Odense, Denmark) with a pre-melt hold of 95 °C for 60 s followed by a hold at 40 °C for 60 s and a melt from 40 °C to 95 °C with a ramping rate of 0.5 °C/s, acquiring fluorescence at each 0.5 °C.

### Design of EpiDirect® MGMT Methylation qPCR Assay
The assay was designed to analyze the methylation status of the four CpG sites used in the melt studies (*MGMT* promoter CpG sites 75 to 78). The oligonucleotide sequences used in the assay can be found in Table 2. A forward EpiPrimer™ and a standard reverse primer amplified a region of 97 base pairs from 129,467,250 to 129,467,346. The EpiPrimer™ contained IPN molecule 2 at each of the CpG sites that it covers. A HydrolEasy™ probe (an INA® based hydrolysis probe) was designed downstream to the reverse primer. The probe had a FAM™ fluorophore placed in the 5′-end and a BHQ®−1 in the 3′-end. The EpiPrimer™ contained one mismatch to the target sequence at the third

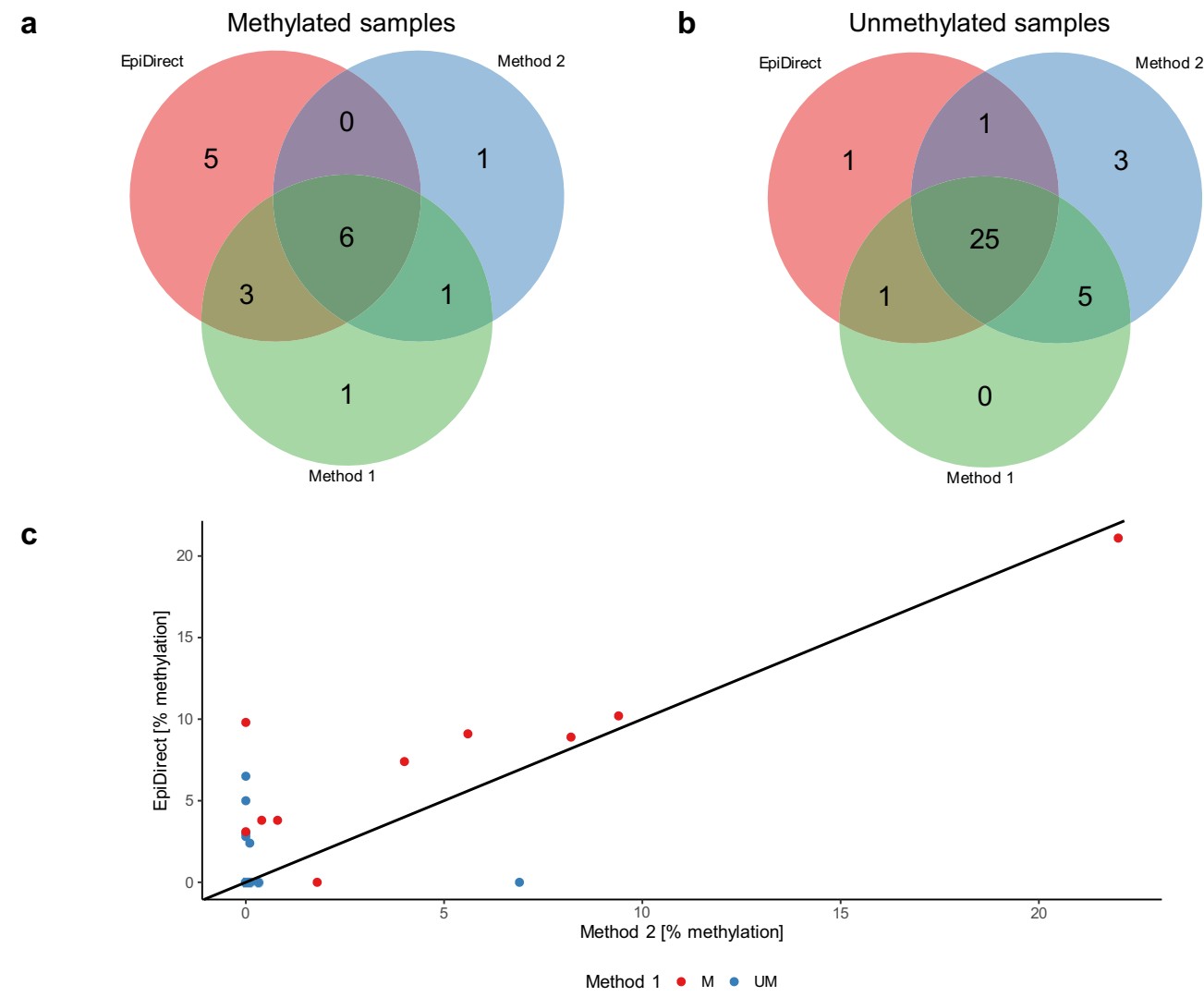

**Fig. 6 | Validation data for the EpiDirect® MGMT Methylation qPCR Assay.**
**a** Venn diagram for samples categorized as methylated by either EpiDirect® MGMT Methylation qPCR Assay (EpiDirect), comparator method 1 (Method 1), and/or comparator method 2 (Method 2). **b** Venn diagram for samples categorized as unmethylated by either of the three methods. **c** Scatter plot of all 42 samples with the percentage methylation calling of EpiDirect plotted against the percentage methylation calling of Method 2. The dots are colored according to the methylation status of Method 1. An identity line ($x = y$) is plotted as a black solid line. Source data are provided as a Source Data file for Fig. 6c.

## Table 1 | Probes and target sequences used for the analysis of DNA and INA® affinities towards unmodified and 5mC modified target sequences

| Sequence ID | Name | Sequence 5' to 3'-end |
|---|---|---|
| P1 | Ref | FAM-CGTCCCGACGCCCG-BHQ1 |
| P2 | INA-1 | FAM-C**1**GTCCC**1**GAC**1**GCCC**1**G-BHQ1 |
| P3 | INA-2 | FAM-C**2**GTCCC**2**GAC**2**GCCC**2**G-BHQ1 |
| T1 | UM | CGGGCGTCGGGACG |
| T2 | M | **mC**GGG**mC**GT**mC**GGGA**mC**G |
| T3 | 1mC | **mC**GGGCGTCGGGACG |
| T4 | 2mC | CGGG**mC**GTCGGGACG |
| T5 | 3mC | CGGGCGT**mC**GGGACG |
| T6 | 4mC | CGGGCGTCGGGA**mC**G |
| T7 | 1,2mC | **mC**GGG**mC**GTCGGGACG |
| T8 | 1,2,3mC | **mC**GGG**mC**GT**mC**GGGACG |

C = non-methylated cytosine; **mC** = 5-methylcytosine; 1 = IPN molecule 1; 2 = IPN molecule 2.

nucleotide from the 5′-end to reduce the $T_m$ as the region is very GC-rich.

An internal reference gene was designed in the housekeeping TATA-Box Binding Protein (*TBP*) gene amplifying a region of 106 base pairs located on chromosome 6, GRCh38.p14 from 170,570,225 to 170,570,330, of which amplification is independent of the methylation status of *MGMT*. The forward and reverse primers were designed to have one mismatch to the target to reduce primer-dimer formation. The reverse primer was designed as a SuPrimer™ (a primer utilizing INA® technology) to increase affinity for the target sequence. As for the *MGMT*, a HydrolEasy™ probe was designed between the primer pairs, with a HEX™ fluorophore in the 5′-end and a BHQ®-1 in the 3′-end.

### PCR set-up
A four times concentrated mix (denoted as 4X) of the primers and probes listed in Table 2 was prepared (denoted as PP mix). The final concentration of each oligo in the PCR tube is also found in Table 2. Ten µL of 2X Ampliqueen master mix (PentaBase A/S) was mixed with 5 µL of the 4X PP mix and 5 µL purified DNA for each PCR set-up. The

**Table 2 | Sequence and working concentration of primers and probes used in the EpiDirect® MGMT Methylation qPCR Assay**

| Gene | Oligo type | Sequence 5' to 3'-end | IPN[a] | Working conc. [nM] |
|---|---|---|---|---|
| MGMT | Forward primer | CGACCCGACGCCCGAGCGCTTCACTGAGACA GGTCCTCGC | 4 | 900 |
| MGMT | Reverse primer | CGAGGGAGAGCTCCGCACTCTTCCG | 0 | 900 |
| MGMT | Hydrolysis probe | FAM AGGCGACCCAGACACTCACCAAG BHQ1 | 4 | 500 |
| TBP | Forward primer | AGGCAGCCATGCCCACCTCACTGC | 0 | 400 |
| TBP | Reverse primer | AGGTCAGGAGGAACCAAGTGAGCCCCA | 1 | 400 |
| TBP | Hydrolysis probe | HEX CACACAGAACTAATGTGCCTGTGAACAG ACACCA BHQ1 | 6 | 400 |

[a]Number of IPN molecules in the sequence. The location of the molecules in the sequence is not shown.

PCR was performed using a CFX Opus 96 Real-time PCR Instrument (Bio-Rad Laboratories, Inc., Hercules, CA, US). The PCR program started with 120 s of hold at 95 °C for activation of the DNA polymerase followed by five pre-cycles with 60 s of denaturation at 100 °C and 60 s of combined annealing and elongation step at 76 °C. This was followed by 40 cycles of 10 s at 98 °C and 60 s at 76 °C. Fluorescence was acquired on the FAM™ and HEX™ channels in the second step at 76 °C.

For the verification experiments, we used Human HCT116 DKO Methylated DNA cat# D5014 (Zymo Research, Irvine, CA, US) and unmethylated Human Genomic DNA cat# G3041 (Promega, Madison, WI, US), referred to as fully methylated and unmethylated DNA, respectively. A no-template control (NTC), 5 ng positive control (25% methylated DNA made by mixing the fully methylated and unmethylated DNA), and 5 ng negative control (unmethylated DNA) were included in each PCR set-up.

### Assay verification

Two linear dynamic range studies were performed to verify the performance of the assay. The first one was on the methylation calling and the second was on the concentration of the DNA. Five replicates were evaluated for each dilution. For the methylation calling, a dilution row was made containing 100%, 50%, 25%, 10%, 5%, and 2.5% methylated DNA. The samples were diluted to a concentration of 1 ng/μL in TE-buffer (10 mM TRIS, 1 mM EDTA). The experiment was repeated to obtain data from two separate PCR runs. Data from these two PCR runs were used to make an equation for the calculation of the methylation percentage based on the ΔCt value. The ΔCt values were plotted against the common logarithm to the percent methylation, and linear regression was made. For the DNA concentration study, a dilution row was made using a factor ten dilution from 10 to 0.01 ng/μL using 25% methylated DNA, corresponding to the positive control. The ΔCt values were plotted against the common logarithm to the DNA concentration, and linear regression was made. The slope of the linear fit was used to calculate the PCR efficiency (E).

The limit of blank (LOB) was evaluated using twenty replicates of unmethylated DNA diluted to 1 ng/μL in TE-buffer, and the technical limit of detection (LOD) was calculated based on the LOB. The LOD was defined as the percentage of methylation where a minimum of 95% of the replicates were detected. The LOD was experimentally assessed using twenty replicates of 3% methylated DNA in a concentration of 1 ng/μL using TE-buffer for dilution. Twenty replicates of unmethylated DNA purified from formalin-fixed paraffin-embedded (FFPE) material from a glioblastoma tumor were analyzed on the EpiDirect® MGMT Methylation qPCR Assay to verify that all replicates would be called unmethylated by the assay based on the LOD ΔCt cut-off value. Twenty replicates of 3% methylated DNA were made by mixing fully methylated DNA with a purified unmethylated FFPE sample to verify the LOD in FFPE material.

### Validation cohort

All patients included in this validation cohort were selected from the database of the Oncology Institute of Southern Switzerland, IOSI, EOC (Switzerland) from 2004 to 2021. All samples were anonymized. The

**Table 3 | Patient characteristics of the 42 patients included in the study**

| | n (%) |
|---|---|
| **Sex** | |
| Male | 24 (57.1) |
| Female | 18 (42.9) |
| **Age** | |
| ≥60 | 25 (59.5) |
| <60 | 17 (40.5) |
| **Diagnosis** | |
| Glioblastoma | 35 (83.3) |
| Anaplastic Astrocytoma | 1 (2.4) |
| Oligodendroglioma | 2 (4.8) |
| Diffuse Astrocytoma | 2 (4.8) |
| Pilocytic Astrocytoma | 2 (4.8) |
| **IDH1 status** | |
| Mutated | 5 (11.9) |
| Wildtype | 37 (88.1) |

tumor sections used for the study were left-over FFPE samples. Inclusion criteria were: Age ≥18 years, histology-proven brain tumor (glioblastoma, oligodendroglioma, or astrocytoma), and availability of leftover tumor tissue or DNA for the validation of the assay. Exclusion criteria were: Age <18 years and insufficient amount of leftover tissue material or extracted DNA.

For the validation cohort, 42 out of 50 tumor samples contained sufficient material for performing all three analyses and these were used for the comparison. The cohort contained primarily glioblastoma samples (83%) and 57.1% of the samples were from men. The age at the time of biopsy spanned from 25 to 79 years with an average of 60.2 years. The data for the cohort of 42 patients are summarized in Table 3.

### Comparator methods

Two methods for the analysis of *MGMT* promoter methylation status were compared to the EpiDirect® Methylation qPCR Assay using the validation cohort. The first method (comparator method 1) was a methyl-specific PCR assay followed by gel electrophoresis designed by and used in clinical routine at Institute of Pathology of Locarno, EOC (Switzerland). The second method (comparator method 2) was a quantitative real-time MSP Assay named geneMAP™ MGMT Methylation Analysis Kit (Genmark Sağlık Ürünleri, Istanbul, Turkey).

For each patient, FFPE tumor tissue was analyzed for quality and tumor content. Genomic DNA was extracted from three 8 μm-thick serial sections of each FFPE block using the QIAamp DNA FFPE tissue kit (Qiagen, Chatsworth, CA, USA) according to the manufacturer's instructions. For the two comparator methods, the purified DNA was treated with bisulfite prior to MSP analysis. The concentration of the DNA was quantified by Nanodrop 1000 (Witec, Littau, Switzerland).

For comparator method 1, around 500 ng of DNA was bisulfite treated using EZ DNA Methylation-Gold cat# D5005 (Zymo Research). Six μL of purified and bisulfite-treated DNA was amplified through MSP in duplicates using MGMT methylated (MGMT-M) or unmethylated (MGMT-UM) specific primers (Supplementary Table 2) and applying the following protocol: 50 °C for 2 min, 95 °C for 5 min, 40 cycles of 60 s of denaturation at 94 °C, annealing at 55 °C (MGMT-M)/57 °C (MGMT-UM) for 60 s and elongation at 72 °C for 60 s. This was followed by 10 min at 72 °C and a hold at 10 °C. A sample was identified as methylated or unmethylated based on the presence of amplification products visualized by running the PCR products on a 3% agarose gel and comparing the cases with positive and negative controls.

For comparator method 2, 200 ng of DNA was converted using EZ DNA Methylation-Lightning Kit cat# D5030 (Zymo Research). Five μL of converted material was analyzed using the geneMAP™ MGMT Methylation Analysis Kit following the manufacturer's instructions. The analysis was performed on a CFX Opus 96 Real-time PCR Instrument (Bio-Rad Laboratories) or CFX96™ (Bio-Rad Laboratories). A cut-off of >0.6% was used according to the manufacturer's guidelines.

The samples were analyzed with EpiDirect® MGMT Methylation qPCR Assay by adding 5 μL of purified DNA (concentration 0.1–10 ng/μL) from each patient to the assay. The DNA was amplified according to the PCR program mentioned previously in the method section using the CFX Opus 96 Real-time PCR Instrument (Bio-Rad Laboratories) or the CFX96™ (Bio-Rad Laboratories). The data were analyzed according to the ΔCt between the FAM™ and HEX™ channel and converted to an estimate of the percentage methylation by the equation obtained from the assay verification studies. The technical LOD was used as a cut-off for calling the samples methylated.

## Statistics and reproducibility
All data analysis and statistics were made in R studio version 1.3.1093 (R version 4.0.3). The following R packages were used; cutpointr (1.1.2), dplyr (1.1.0), ggplot2 (3.4.1), ggmisc (0.5.2), ggpubr (0.6.0), readxl (1.4.2), tidyr (1.3.0), VennDiagram (1.7.3). The $T_m$ of the DNA duplexes was determined by the maximum value of the first negative derivative of the melt curves. This was done automatically in the PCR software (Real-time PCR system V1) belonging to the BaseTyper48.4 Quiet HRM Real-Time PCR System (PentaBase A/S). Mean $T_m$ values were reported with standard deviation denoted by ±. Differences in $T_m$ values were calculated with 95% confidence intervals assuming the technical replicates followed a normal distribution.

The PCR cycle threshold (Ct) values were determined in the CFX Maestro Software 2.0 (Bio-Rad Laboratories). The thresholds were manually set for both the FAM™ and HEX™ channels according to 10% of the maximum relative fluorescence units (RFU) of the positive control sample (25% methylated DNA). The ΔCt between the two channels was calculated by Eq. 2.

$$\Delta Ct = Ct_{FAM} - Ct_{HEX} \tag{2}$$

The PCR efficiency was calculated by Eq. 3.

$$E = 10^{\frac{-1}{slope}} - 1 \tag{3}$$

A significance level of 0.001 was chosen, giving a critical ($Z_\alpha$) value of 2.576 for the LOB study. The mean value ($\mu_B$) of the unmethylated replicates and the standard deviation ($\sigma_B$) was used to calculate the LOB using Eq. 4.

$$LOB_{\Delta Ct} = \mu_B - Z_\alpha \cdot \sigma_B \tag{4}$$

The LOD at 95% certainty was calculated from the LOB using Eq. 5 with 19 degrees of freedom ($f$), as 20 replicates were used for the study.

$$LOD_{\Delta Ct} = LOB_{\Delta Ct} + \frac{1.645}{1 - \left(\frac{1}{4} \cdot f\right)} \cdot \sigma_B \tag{5}$$

The sensitivity and specificity of the assay were calculated by Eq. 6 and Eq. 7, respectively. The 95% confidence intervals were calculated using the Wilson score.

$$Sensitivity[\%] = \frac{True\ Positive}{True\ Positive + False\ Negative} \cdot 100 \tag{6}$$

$$Specificity[\%] = \frac{True\ Negative}{True\ Negative + False\ Positive} \cdot 100 \tag{7}$$

For the validation study, no statistical method was used to pre-determine the sample size. Eight samples were excluded from the study due to insufficient amount of sample material. The investigators were blinded to the methylation status of comparator method 1 when performing the analysis and data collection of comparator method 2 and EpiDirect® MGMT Methylation qPCR Assay. The final comparison with the methylation status evaluated using the three methods was done at the end, after the analysis of comparator methods 1 and 2.

## Reporting summary
Further information on research design is available in the Nature Portfolio Reporting Summary linked to this article.

## Data availability
The raw patient sample characteristics data are protected and are not available due to data privacy laws. The data generated in this study are provided in the Supplementary Information and in the Source Data file. Source data are provided with this paper.

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

## Acknowledgements

We disclose support for the research of this work from Innovation Fund Denmark's Industrial PhD Program [grant number 1044-00027B] (K.K.B.) and Eureka Eurostars [grant number E! 12513] (K.K.B., M.M., U.B.C., R.K.P., and M.F.). Figures 1, 3a, and 4a were created using BioRender.com with publication rights.

## Author contributions

K.K.B. was responsible overall for preparation of the manuscript and developing the technology presented in the paper. M.M. assisted in development of the technology and was responsible for the assay verification. S.E. performed the experimental part on patients and helped in the preparation of the manuscript. G.D. performed the experimental part on patients. P.S. revised all the histological material and identified the regions on which DNA extraction had to be performed. F.M. performed patients' selection and resection. He helped in the preparation and submission of the documents to the local Ethics Committee. E.A.C., M.S. and R.K.P. have been supervisors for the academic projects leading to this manuscript. U.B.C. and R.K.P. have conceptualized the idea of the technology. M.F. planned the entire work on patients. He evaluated the results, prepared, and submitted the documents to the Ethics Committee. All authors have read and approved the final version of the manuscript.

## Competing interests
K.K.B., M.M., R.K.P. and U.B.C. were employed at PentaBase A/S during the study who is a manufacturer and seller of EpiDirect® MGMT Methylation qPCR Assay utilized in this study. U.B.C. and R.K.P. are co-owners of PentaBase A/S. K.K.B., U.B.C. and R.K.P. are all founders of a patent application covering the technology utilized in this study [International Publication number: WO2022002958]. The remaining authors declare no competing interests.
