## [Peer Review File · Nature Communications]

REVIEWER COMMENTS

Reviewer #1 (Remarks to the Author):

The manuscript is well-written, and the text is presented in a clear and stringent way. The figures illustrate the novel technology in a clear and instructive way.

Introduction

Part of the text in the introduction is too general, as the sentence: Cancer cells (line 28). This is a mixture of locus-specific methylation changes affecting TSGs and genome-wide hypomethylation, the latter is not a target of this article - describing a locus-specific methylation detection method. It may be more relevant to describe the challenges in analyzing clinically relevant methylation-specific biomarkers in patient tissues. The degradation of DNA (FFPE tissue) limited available DNA (liquid biopsies), which after bisulfite conversion, lowered the yield further.

It may be of interest to the reader to have a description of bisulfite conversion and the challenges as well as the methylation detection methods relying on pretreatment by bisulfite conversion. Why are only MSP and pyrosequencing mentioned? It would be logical to compare this new technology to other relying on qPCR and melting of the amplicons too. Different technologies could be listed overall in groups depending on the underlying technologies.

The references on comparison of bis-treatment kits are a bit old, the quality of these kits has improved over the past years.

The description of the intercalating Nucleic Acid (INA) is explained logically and in detail, as how the base-stacking stabilizes the DNA double-helix. This further clarifies how the implementation of intercalating pseudo-nucleotides (IPN) affects the π -stacking energy and thereby the T_m of the double-stranded DNA. Two very informative sections.

Results

The EpiPrimer™ is well-described and in detail with clear figures, so it is easy to picture how it works. In the following section, the nature of the reference sequence could be explained in brief (line 94). Also, it is detected by an independent fluorescent channel, so which fluorescent dye is used? As this reference

sequence is very important for the understanding of the process and the results, it should be explained in a bit more detail in this section.

Each step of the validation of the EpiDirect® technology is thoroughly described and accompanied by well-composed figures. The analyses of 20 replicates of unmethylated and 3% methylated DNA provide highly convincing results.

In two commercially available MGMT methylation detection methods, the results from 42 patient samples were compared to the EpiDirect® technology. The sensitivity and specificity between one method and EpiDirect® were 0.82 and 0.84, respectively. Compared with the second method the sensitivity was 0.75 and the specificity 0.76, respectively. Five samples were detected as methylated by EpiDirect® and as unmethylated by the other methods. These five samples had a methylation level ranging from 2.4%-6.5% as calculated by the EpiDirect® algorithm. This is important in the context of analyzing clinical samples for MGMT methylation in order to determine the treatment regime for instance glioblastoma patients. Recent meta-studies discuss the cut-off threshold of MGMT methylation on glioblastoma patient samples and point to the importance of being able to detect any MGMT methylation. In this light, the new EpiDirect® platform is very interesting.

Further questions

Can heterogeneous methylation be detected by EpiDirect®?

Is there a limitation on the length of the anchor sequence?

Is the EpiPrimer™ technology available for other target genes?

Is the INAs® and EpiPrimer™ technology available only through Pentabase?

Conclusion

The EpiDirect® platform presented here is highly interesting and relevant when analyzing DNA methylation at specific genomic regions. Avoiding bisulfite-conversion is not only reducing hands-on time but also preserves the integrity and amount of DNA. These are important aspects especially when analyzing clinical DNA samples from archive FFPE tissue or from liquid biopsies.

The EpiPrimer™ technology is novel and well-thought. All steps of the method validation are sound and meet the expected standards of this field. The details of the procedure are sufficient to reproduce the work.

The technology is compared to two commercially available DNA methylation tests for the MGMT gene, both relying on bisulfite conversion prior to the methylation analysis. Here, the EpiDirect® platform is able to detect low-level methylation of five clinical samples, which are categorized as unmethylated by the other methods. The EpiDirect® platform holds the promise of a sensitive and robust technology avoiding the bisulfite-conversion step to assess DNA methylation. The

The EpiDirect® platform with the EpiPrimer™ design is highly interesting and relevant to all working in the field of measuring DNA methylation.

Reviewer #2 (Remarks to the Author):

Overview

The manuscript by Bendixen et al. presents a novel qPCR-based technology, named EpiDirect®, which allows for direct quantification of methylation levels using untreated DNA. EpiDirect® is based on the previously described intercalating nucleic acid (INA®) platform, which enables users to distinguish between methylated and unmethylated cytosines in an analyzed sequence with a special primer design. With an assay that was developed based on this technology, the authors analyze the methylation status of the MGMT gene promoter, which is used as a predictive biomarker for effectiveness of temozolomide treatment in patients with gliomas. The utility of the assay is further compared with two bisulfite-based methylation-specific PCR assays.

Relevance

Assessment of DNA methylation cannot be performed using standard PCR methods, due to the loss of methylation patterns during the amplification process. The most common method for overcoming this challenge is the bisulfite treatment of DNA, leading to a permanent conversion of DNA, in which unmethylated cytosines are changed to uracils while methylated cytosines remain unchanged. This method, however, is time-consuming and harmful to DNA, which is emphasized by the authors. The conversion of the DNA sequence also makes primer design challenging. Alternatives to bisulfite treatment include methods involving the enzymatic treatment of DNA. These methods, however, are sequence-dependent. Conversely, the technology presented by the authors allows for the quantification of methylation levels in untreated DNA and therefore may save valuable time in laboratory practice while decreasing the risk of possible laboratory errors. The manuscript is well-written and clearly describes the authors' analysis of the sensitivity, specificity, and utility of the technology. Overall, this is a great contribution to the field, despite the criticism added below.

Major comments:

1. The authors show the utility of the assay developed based on EpiDirect® technology to assess methylation status of the MGMT gene promoter in glioma patients. This analysis is justified, as determination of MGMT gene promoter methylation status is a well-known, clinically significant assessment. However, the authors do not indicate the possibility of using the technology in the assessment of the methylation status of any other genes, especially those for which designing primers while using a bisulfite converted template would be challenging.

2. The utility of the assay for MGMT methylation assessment is only compared to methylation-specific PCR assays. Comparison to other assays based on e.g. pyrosequencing (therascreen® MGMT Pyro® Kit, QIAGEN GmbH, Germany) or MS-HRM (EpiMelt MGMT methylation detection kit, MethylDetect ApS, Denmark) technologies would significantly increase the value of the performed analysis.

Minor comments:

Line 13 In this part of the sentence: “and is involved in development and progression”, “is involved in” should be replaced by e.g. “alterations in DNA methylation are involved in the”, which conveys that DNA methylation is a physiological mechanism, and changes in this mechanism leads to cancer development

Line 15 Preposition “of” should be changed to “on”

Line 107 The clarity of Figure 1 would be improved with the addition of labels to specific parts of the Figure, which could then be referred to in the Figure description

Line 156 and 158 “methyl-group” should be changed to “methyl group”

Reviewer #3 (Remarks to the Author):

A novel qPCR technology for direct quantification of methylation in untreated DNA

The authors describe a novel PCR-based method, EpiDirect, for the quantitative determination of the level of DNA methylation in an amplicon of interest. Methylation of MGMT is given as an exemplar with hypermethylation of this gene being relevant to treatment of brain tumours. Commercial tests exist for the determination of MGMT methylation via traditional methods and the authors compare the performance of the MGMT EpiDetect assay to two of these.

* Comments *

L50-57. In addition to TAPS, mention the New England Biosciences EM-seq method and compare this to bisulphite conversion. EM-seq is increasingly prevalent whereas TAPS is not readily available in kit form as the IP spin out, Base Genomics, was purchased by Exact Sciences.

Fig 4A. To assist interpretation for the reader, please title the two cartoons as 'M' and 'U'.

L202-218. It is not obvious from the text here if the EpiDirect assay was on DNA from FFPE samples or fresh frozen?

L215-218. Please also compare the EpiDirect assay to a combination reference. So, only include the samples in Fig 6 where the comparison Methods 1 and 2 are in agreement and remove the samples in disagreement. This approximation of the EpiDetect assays false negative rate is useful. Also comment on this combination reference across L257-262.

L238. Replace the word "notice" with "observation".

L277. Comment on the recommendations for EpiDirect. Is it useful for FFPE sections, or do you recommend DNA from fresh frozen samples?

L334. Table 2. The reverse primer for the EpiDirect assay will bind across from a CG site. Have you examined whether this primer is influenced by the methylation state of the template DNA?

Response to reviewers

Reviewer 1

Comment	Response
Overall	
The manuscript is well-written, and the text is presented in a clear and stringent way. The figures illustrate the novel technology in a clear and instructive way.	Thank you very much for this comment
Introduction	
Part of the text in the introduction is too general, as the sentence: Cancer cells (line 28). This is a mixture of locus-specific methylation changes affecting TSGs and genome-wide hypomethylation, the latter is not a target of this article - describing a locus-specific methylation detection method. It may be more relevant to describe the challenges in analyzing clinically relevant methylation-specific biomarkers in patient tissues. The degradation of DNA (FFPE tissue) limited available DNA (liquid biopsies), which after bisulfite conversion, lowered the yield further.	Thank you for this suggestion. We have deleted the sentence starting with “Cancer cells...”. As you have suggested we have elaborated a bit on the clinical challenges with analysis of methylation-specific biomarkers in the end of section two of the introduction.
It may be of interest to the reader to have a description of bisulfite conversion and the challenges as well as the methylation detection methods relying on pretreatment by bisulfite conversion. Why are only MSP and pyrosequencing mentioned? It would be logical to compare this new technology to other relying on qPCR and melting of the amplicons too. Different technologies could be listed overall in groups depending on the underlying technologies.	We have added a short section about qMSP and MS-HRM following the MSP and pyrosequencing to have a more exhaustive description of the methods available in the literature.
The references on comparison of bis-treatment kits are a bit old, the quality of these kits has improved over the past years.	We see your point in this observation. We have added a newer paper regarding performance evaluation of different bisulfite conversion kits from 2021.
The description of the intercalating Nucleic Acid (INA) is explained logically and in detail, as how the base-stacking stabilizes the DNA double-helix. This further clarifies how the implementation of intercalating pseudo-nucleotides (IPN) affects the π -stacking energy and thereby the T_m of the double-stranded DNA. Two very informative sections.	Thank you for this comment
Results	
The EpiPrimer™ is well-described and in detail with clear figures, so it is easy to picture how it works. In the following section, the nature of the reference sequence could be explained in brief (line 94). Also, it is detected by an independent fluorescent channel, so which fluorescent dye is used? As this reference sequence is very important	Thank you for this consideration. This section is meant as a general explanation of the technology. We do not believe that reference sequence is relevant for the technology. We could in principle have used any reference gene not prone to duplications, and the dye used could also be changed. We have chosen to give the details regarding the dyes and sequences in the

for the understanding of the process and the results, it should be explained in a bit more detail in this section.	method section. We have added “The reference assay is designed in a gene independent of methylation status.” to the third section of “Novel platform for direct methylation quantification”.
Each step of the validation of the EpiDirect® technology is thoroughly described and accompanied by well-composed figures. The analyses of 20 replicates of unmethylated and 3% methylated DNA provide highly convincing results.	Thank you for this comment
In two commercially available MGMT methylation detection methods, the results from 42 patient samples were compared to the EpiDirect® technology. The sensitivity and specificity between one method and EpiDirect® were 0.82 and 0.84, respectively. Compared with the second method the sensitivity was 0.75 and the specificity 0.76, respectively. Five samples were detected as methylated by EpiDirect® and as unmethylated by the other methods. These five samples had a methylation level ranging from 2.4%-6.5% as calculated by the EpiDirect® algorithm. This is important in the context of analyzing clinical samples for MGMT methylation in order to determine the treatment regime for instance glioblastoma patients. Recent meta-studies discuss the cut-off threshold of MGMT methylation on glioblastoma patient samples and point to the importance of being able to detect any MGMT methylation. In this light, the new EpiDirect® platform is very interesting.	Thank you for this comment, we are aiming to make more clinical studies to investigate the optimal cut-off.
Further questions	
Can heterogeneous methylation be detected by EpiDirect®?	Due to the high sensitivity of our technology, we should be able to identify methylation even in heterogeneous samples. However, it would be nearly not possible to distinguish these cases from those with few tumor cells methylated in a context of a several normal cells (for example, in situations where tumor cells are dispersed, and scraping is a challenge).
Is there a limitation on the length of the anchor sequence?	In the second section of the discussion, we have discussed that that anchor sequence is 14 nucleotides long, and if it was longer it would result in a very high annealing temperature of the PCR. So, there are some limitations to the anchor sequence.
Is the EpiPrimer™ technology available for other target genes?	We have added this to the last section of the discussion. The technology can be transferred to other target genes. However, we do not have any products ready for sale at this point.
Is the INAs® and EpiPrimer™ technology available only through Pentabase?	Yes, the EpiPrimer™ is a patented technology. PentaBase’s patent on INA® is expired, however the use of the INA® technology is patented in several different applications.
Conclusion	

The EpiDirect® platform presented here is highly interesting and relevant when analyzing DNA methylation at specific genomic regions. Avoiding bisulfite-conversion is not only reducing hands-on time but also preserves the integrity and amount of DNA. These are important aspects especially when analyzing clinical DNA samples from archive FFPE tissue or from liquid biopsies. The EpiPrimer™ technology is novel and well-thought. All steps of the method validation are sound and meet the expected standards of this field. The details of the procedure are sufficient to reproduce the work. The technology is compared to two commercially available DNA methylation tests for the MGMT gene, both relying on bisulfite conversion prior to the methylation analysis. Here, the EpiDirect® platform is able to detect low-level methylation of five clinical samples, which are categorized as unmethylated by the other methods. The EpiDirect® platform holds the promise of a sensitive and robust technology avoiding the bisulfite-conversion step to assess DNA methylation. The The EpiDirect® platform with the EpiPrimer™ design is highly interesting and relevant to all working in the field of measuring DNA methylation.	We thank the reviewer for the overall positive evaluation of our technology and for the comments to improve the quality of our manuscript.
--	---

Reviewer 2

Comment	Response
Overview	
The manuscript by Bendixen et al. presents a novel qPCR-based technology, named EpiDirect®, which allows for direct quantification of methylation levels using untreated DNA. EpiDirect® is based on the previously described intercalating nucleic acid (INA®) platform, which enables users to distinguish between methylated and unmethylated cytosines in an analyzed sequence with a special primer design. With an assay that was developed based on this technology, the authors analyze the methylation status of the MGMT gene promoter, which is used as a predictive biomarker for effectiveness of temozolomide treatment in patients with gliomas. The utility of the assay is further compared with two bisulfite-based methylation-specific PCR assays.	Thank you for this comment
Relevance	
Assessment of DNA methylation cannot be performed using standard PCR methods, due to the loss of methylation patterns during the amplification process. The most common method for overcoming this challenge is the bisulfite treatment of DNA, leading to a permanent conversion of DNA, in which unmethylated cytosines are changed to uracils while methylated cytosines remain unchanged. This method, however, is time-consuming and harmful to DNA, which is emphasized by the authors. The conversion of the DNA sequence also makes primer design challenging. Alternatives to bisulfite treatment include methods involving the enzymatic treatment of DNA. These methods, however, are sequence-dependent. Conversely, the technology presented by the authors allows for the quantification of methylation levels in untreated DNA and therefore may save valuable time in laboratory practice while decreasing the risk of possible laboratory errors. The manuscript is well-written and clearly describes the authors' analysis of the sensitivity, specificity, and utility of the technology. Overall, this is a great contribution to the field, despite the criticism added below.	We really thank the reviewer for the positive overall evaluation of our technology.
Major comments	
1. The authors show the utility of the assay developed based on EpiDirect® technology to assess methylation status of the MGMT gene promoter in glioma patients. This analysis is justified, as determination of MGMT gene	We have added this to the last section of the discussion. The technology can be transferred to other target genes, however, we do not have any products ready for sale at this point.

promoter methylation status is a well-known, clinically significant assessment. However, the authors do not indicate the possibility of using the technology in the assessment of the methylation status of any other genes, especially those for which designing primers while using a bisulfite converted template would be challenging.	
2. The utility of the assay for MGMT methylation assessment is only compared to methylation-specific PCR assays. Comparison to other assays based on e.g. pyrosequencing (therascreen® MGMT Pyro® Kit, QIAGEN GmbH, Germany) or MS-HRM (EpiMelt MGMT methylation detection kit, MethylDetect ApS, Denmark) technologies would significantly increase the value of the performed analysis.	We completely agree with your comment. However, it has taken several months to get approval for samples analyzed by pyrosequencing, so it has not been possible to include in this paper. In this paper, we have focused on the technical features of the EpiDirect MGMT assay. We will follow-up with other studies which will have a more clinical focus and comparison to therascreen® MGMT Pyro® Kit, QIAGEN GmbH, Germany with larger cohorts (>100 samples). This paper aims to describe the method and to prove its clinical potential using a smaller cohort of samples.
Minor comments	
Line 13 In this part of the sentence: “and is involved in development and progression”, “is involved in” should be replaced by e.g. “alterations in DNA methylation are involved in the”, which conveys that DNA methylation is a physiological mechanism, and changes in this mechanism leads to cancer development	Thank you for this comment. We have implemented your suggestion.
Line 15 Preposition “of” should be changed to “on”	Thank you for this comment. We have implemented your suggestion.
Line 107 The clarity of Figure 1 would be improved with the addition of labels to specific parts of the Figure, which could then be referred to in the Figure description	We have added a label for the reverse primer, we hope that this helps with clarity, and divided the figure into “a” and “b”. Please elaborate which labels to add if this is not sufficient.
Line 156 and 158 “methyl-group” should be changed to “methyl group”	Thank you for this comment. We have implemented your suggestion.

Reviewer 3

Comment	Response
Overall	
The authors describe a novel PCR-based method, EpiDirect, for the quantitative determination of the level of DNA methylation in an amplicon of interest. Methylation of MGMT is given as an exemplar with hypermethylation of this gene being relevant to treatment of brain tumours. Commercial tests exist for the determination of MGMT methylation via traditional methods and the authors compare the performance of the MGMT EpiDetect assay to two of these.	Thank you for this comment
Comments	
L50-57. In addition to TAPS, mention the New England Biosciences EM-seq method and compare this to bisulphite conversion. EM-seq is increasingly prevalent whereas TAPS is not readily available in kit form as the IP spin out, Base Genomics, was purchased by Exact Sciences.	We have implemented your suggestion about EM-seq right after we have mentioned TAPS.
Fig 4A. To assist interpretation for the reader, please title the two cartoons as ‘M’ and ‘U’.	We have added this to both Fig 4A and 3A.
L202-218. It is not obvious from the text here if the EpiDirect assay was on DNA from FFPE samples or fresh frozen?	The samples are FFPE. We have added this to first section of “EpiDirect® MGMT Methylation qPCR Assay validation”.
L215-218. Please also compare the EpiDirect assay to a combination reference. So, only include the samples in Fig 6 where the comparison Methods 1 and 2 are in agreement and remove the samples in disagreement. This approximation of the EpiDetect assays false negative rate is useful. Also comment on this combination reference across L257-262.	Thank you for this comment. We have calculated the sensitivity and specificity using the combination reference, and added comments to this in the discussion as you suggest.
L238. Replace the word “notice” with “observation”.	We have replaced this, thank you.
L277. Comment on the recommendations for EpiDirect. Is it useful for FFPE sections, or do you recommend DNA from fresh frozen samples?	The method is only validated on FFPE sections, but we have no reason to believe that it cannot work with fresh frozen samples. Sorry, we do not agree with adding a sample recommendation as the final part of the discussion. We have an instruction for use for this assay, which can be shared upon request.
L334. Table 2. The reverse primer for the EpiDirect assay will bind across from a CG site. Have you examined whether this primer is influenced by the methylation state of the template DNA	We thank the reviewer for the precision of his/her revision. During the development, we tested this by replacing the EpiPrimer with a regular primer. We tested 100% methylated DNA and unmethylated DNA which we have determined the concentration of on another qPCR assay, and we saw that the Ct value of the 100% methylated DNA and the unmethylated DNA were the same, meaning no bias in the PCR of methylated vs unmethylated DNA by the reverse primer.

REVIEWERS' COMMENTS

Reviewer #1 (Remarks to the Author):

The authors have addressed the concerns and answered all questions to full satisfaction.

If possible, reference 10 should be replaced or supplemented with an article explaining the development of a methylation-sensitive high-resolution method. Ref. 10 describes the optimization of a “High-Resolution Melting-based quantitative analysis”.

Reviewer #2 (Remarks to the Author):

The authors have addressed the concerns I have raised

Reviewer #3 (Remarks to the Author):

The authors have sufficiently addressed all my comments. Thank you.

Response to reviewers

Reviewer 1

Comment	Response
The authors have addressed the concerns and answered all questions to full satisfaction. If possible, reference 10 should be replaced or supplemented with an article explaining the development of a methylation-sensitive high-resolution method. Ref. 10 describes the optimization of a “High-Resolution Melting-based quantitative analysis”.	We are happy to hear that you find our response satisfying. We have replaced reference 10 with a reference that describe the development of methylation-sensitive high-resolution melt and a reference describing the development of a qMSP assay.

Reviewer 2

Comment	Response
The authors have addressed the concerns I have raised	We are happy to hear that you find our response satisfying.

Reviewer 3

Comment	Response
The authors have sufficiently addressed all my comments. Thank you.	We are happy to hear that you find our response satisfying.